# Empirical Measurement of Competition in the Thai Banking Industry

**Jirawan Prayoonrattana \*, Thanarak Laosuthi and Bundit Chaivichayachat**

Department of Economics, Kasetsart University, 50 Phahonyothin Rd, Chatuchak, Bangkok 10900, Thailand;
fecotrl@ku.ac.th (T.L.); fecobdc@ku.ac.th (B.C.)

**\*** Correspondence: jirawan.pra2019@gmail.com

**Abstract:** The degree of competition in the banking industry can be observed and measured by two approaches, structural and nonstructural. Based on these two approaches, there are various indicators, which are different factors and methods. This paper aims to provide calculations, determine a good indicator, and assess the competitive environment of the Thai banking industry. Specifically, there are four indicators—concentration ratio, Herfindahl–Hirschman Index, Lerner Index, and Panzar–Rosse H statistic—which are widely used to examine the efficiency and effectiveness of policies in the banking industry. The findings indicate that the Lerner Index, calculated by stochastic frontier analysis, is the most reliable indicator of the banking competition environment in Thailand. It has a range of 0.36 to 0.60 and an average value of 0.40. Furthermore, during the period of study, the degree of Thai banking competition had a tendency to increase over time, which reflects an increase in allocative efficiency of resources in the banking industry. This is in accordance with the Financial Sector Master Plan of the country. However, this result probably leads to instability of the financial system. Therefore, policy-makers should carefully regulate competition policy by considering the systematic risk of the banking system at the same time.

**Keywords:** banking; competition; Lerner Index

## 1. Introduction

The banking sector is one of the most significant industries that has economic importance. Theoretically, banks act as intermediaries in the financial system, which allocate resources (funds) from savers to borrowers. The essential role of banks is to resolve asymmetric information in the financial market. Thailand is a developing country, which has a high degree of asymmetric information in financial market. Thus, all economic activities depends heavily on financial intermediaries, which are banks.

Competition in the Association of Southeast Asian Nations (ASEAN) banking sector (including Thailand) dramatically changed after the Asian financial crisis in 1997–1998. Mergers and acquisitions in the banking industry are critical in order to achieve financial stability in these regions. Moreover, the integration in the banking market is due to the Banking Integration Framework (BIF) in 2020 (Khan et al. 2016). Competition in the financial industry is often claimed as one factor involved in the global financial crisis in 2007–2009 (Coccorese 2014). After the global financial crisis, competition in the banking sector became a worthwhile subject of study for policy-makers and researchers (Chileshe 2017). Therefore, measurement of the degree of competition in the banking sector needs to be updated (Claessens 2009).

Furthermore, the existing literature indicates that role of competition in the banking sector has some special properties that differ from other industries. It is importance for the efficiency of production in banking industry, similar to other industries. At the same time, the stability of the banking system is also crucial for effective supervision (Claessens 2009). In aspect of efficient production in financial

service, level of bank competition has negatively associated with prices of bank's products (Anzoategui et al. 2010). A high competition leads to reduce bank's prices (interest rates). Consequently, this can induce a greater of businesses investment and household consumption, which in turn to foster economic development (Ghosh 2018). In addition to the prices, Ghosh (2018) points out that a high level of asymmetric information problem in developing countries would probably increase the costs of acquiring information of their borrowers. Therefore, the less competition can decrease efficiency in this sector by increasing costs of credit for borrower. Consequently, the less competitive in banking sector also leads to inefficiency allocation of banking products. In aspect of effective policy, competition in this sector can be correlated with government regulation. Since banks act as an intermediates, who use primary source of funds from saving of households and firms for investing in credit market. Especially in Thailand, overall economic activity depends heavily in bank credit. Hence, this sector is under the supervision of the central banks. Beck et al. (2004) point out that regulatory implementation can affect the behavior of bank's competition, which is not depending on the actual market structure. For instance, when the central banks strictly regulate the entry of new player in this sector, it will causes of obstacle to access financial products. Consequently, this may causes to increase monopoly power, reduces the contestability and competitiveness in banking industry. Some studies point out that competition can positively or negatively relates to stability of banking system (Beck 2008; Claessens 2009; Rao Subramaniam et al. 2019). Beside, numerous studies state that competition is one factors that significantly affects conduct of monetary policy transmission through banking sector (Olivero et al. 2011; Fungáčová et al. 2013; Khan et al. 2016). Up to this point, competition is the major focus for understanding banking sector in various aspects which cannot be abandoned.

Even though there are many indicators that are used as proxies to measure competition, most studies use one indicator over another. However, there seems to be no consensus on which indicators are the best measurement. Moreover, different indicators are derived from different factors, which may contribute to dissimilar inferences about the interpretation of competition. Choosing competition indicators is highly significant for the understanding of practical outcomes. Therefore, the advantages and disadvantages of each indicator should be carefully considered.

For the above reasons, the analysis of competition in the banking sector should be improved by studying more indicators as proxies for competition. This study approaches four main indicators: concentration ratio (CR), Herfindahl–Hirschman Index (HHI), Lerner Index (LI), and Panzar–Rosse H statistic (PRH), as the popular methods in economics, finance, banking, and monetary policy. Even though there are many studies in this area, there has been no study proving which indicators are strongly represented to the greatest extent in the Thai banking industry, contributing to an analysis of competition in the banking sector.

To fill this gap, structural and nonstructural approaches as empirical methods are used to determine the degree of competition. As studied in previous research (Bikker and Haaf 2002; Anzoategui et al. 2010, 2012; Fungáčová et al. 2013) using the structural approach on the traditional industrial organization, banking competition should be considered based on market structure, measured by the concentration ratio (CR) or the Herfindahl–Hirschman Index (HHI). For a nonstructural approach, as found in studies on the new empirical industrial organization (NEIO), banking competition should be investigated based on the bank's conduct, which includes the bank's market power (LI) or elasticity of its revenue with respect to variation of its input factors (PRH) (Bikker and Haaf 2002; Anzoategui et al. 2010, 2012; Fungáčová et al. 2013).

The rest of this study is organized as follows: Section 2 presents the structure of the Thai banking sector, Section 3 describes the literature review of banking competition, Section 4 explains the methodological approach and describes the data used to obtain the measurement of competition, Section 5 reports the results of the analysis of banking competition in Thailand, and Section 6 presents conclusions.

## 2. Competition of Banking Industry in Thailand

The banking industry plays an extremely significant role in the Thai economy. The Thai banking industry has largely been dominant because domestic credit to the private sector, which is provided by the banking sector, increased from 93% in 2001 to 112.53% in 2018 (% of gross domestic product (GDP)) (World Bank 2019). Credit from banks is a crucial factor in real sector behaviors. Overall economic activity in the country requires support by the banking industry. Banking business can be classified into four types: domestic commercial banks, retail banks, foreign financial institution representative offices, and specialized financial institutions (SFIs) (Bank of Thailand 2019). Domestic commercial banks act as depository institutions, which do business by accepting deposits subject to withdrawal on demand or the end of maturity date from savers and lending funds to potential bank borrowers. In Thailand, there is only one retail bank, and its business operation focuses specifically on microfinance lending. However, most commercial banks certainly do microfinance lending business as well. A core business of foreign financial institution representative offices is to facilitate their clients who do business in Thailand. SFIs are incorporated for special purposes in order to support fiscal policy implementation. Because of different core business operations, they are not direct competitors. Therefore, domestic commercial banks are emphasized and explained in terms of the competition of the banking sector.

After the Asian crisis in 1998, Thai domestic commercial banks experienced significant reforms. The number of mergers and acquisitions of small banks increased over a period of time. In January 2004, the Financial Sector Master Plan (FSMP) was established in order to assess and strengthen the financial system. Particularly, as indicated by Kubo (2006), one of the most important objectives for setting the FSMP was likely a signal for an increase in environmental competition in the banking industry (Kubo 2006).

During FSMP Phase I (2004–2008), the Bank of Thailand (BOT) issued new bank licenses. Three new banks were established: TISCO bank (TISCO) in 2005,[1] Land and House bank (LH) in 2006,[2] and Thai Credit for Retail bank (TCR) in 2007.[3] The emergence of new banks reflects the ease of access to bank services and fostering of competition. The main aim of FSMP Phase II (2010–2014) and Phase III (2016–2020) was to enhance financial efficiency by promoting competition and enhancing financial access. This resulted in increased competition in the banking industry. Because of the availability of new technology, banking activities were rapidly transformed from bank branches and ATMs to Internet platforms. Obviously, customer services such as mobile banking platforms, prompt pay system, and other digital payments increased. At the beginning, because of the expensive costs and low reliability of the system, especially in security, the use of banking technology in Thailand was unpopular. However, as driven by regulators, transaction fees for using the new technology were removed, and the amount of usage increased. Moreover, the financial technology was indirectly accelerated by Thai governmental policy, which can be called Thailand 4.0. The policy was implemented in 2016 and was aimed at creating innovation, new technology, and high-quality services of many industries. Overall, the change in circumstances induced a competitive environment in this sector. Competition in Thai banking has absolutely changed by relaxing the barriers for new banks and reducing restrictions to accessing financial products.

## 3. Literature Review

The analysis of industrial competition in microeconomics is based on market structure theory. There are two theoretical concepts of industrial competition, static and dynamic views. The static view believes that the long-run equilibrium of industrial competition would exist if the industry is

---

[1]　TISCO upgraded its status from finance company to commercial bank according to the Financial Sector Master Plan of the Bank of Thailand in 2005.

[2]　LH bank was incorporated as a retail bank in 2006 and became a commercial bank in 2011.

[3]　Thai Credit for Retail (TCR) bank was incorporated as a retail bank in 2007.

characterized as a perfect competition market[4] subject to a given of constant technology. Imperfect competition derives from an advantage of production processes, such as economies of scale and lower average costs, which contribute to higher market power of one over its rivals in both price competition and non-price competition. In contrast to the static view, the dynamic view argues that the market is always imperfect. Imperfect competition stems from the latest innovations or product differentiations and the technological progress of production. Moreover, the monopoly status is impermanent due to creative destruction (Lipczynski et al. 2017). However, empirical studies in the banking industry often measure the degree of competition from the static view. This is because the related factors in dynamic view, which are technological progress and innovations of banking firms, are difficult to observe.

At this point in this study, the literature review mainly focuses on empirical approaches from the static view, which have frequently been used to measure competition in the banking industry. Empirical studies are divided into two approaches, structural and nonstructural. An explanation of each approach comprises underlying theory, advantages, limitations of each indicator, and empirical results.

*3.1. Structural Approach*

In the empirical studies, the structural approach is based on the structure–conduct–performance (SCP) paradigm, which can be linked to a relationship between the market structure and the firm's conduct in the market's performance. In this case, the degree of competition can be assessed by the firm's conduct. Structure commonly refers to the market structure, which is measured by many factors: the number and size of buyers and sellers, the entry–exit conditions, or product differentiation. Conduct refers to the behaviors of each firm in the market, such as pricing policies, collusion, mergers, or business objectives. Performance refers to industrial outcomes, such as product quality, profitability, productive efficiency, and allocative efficiency (Lipczynski et al. 2017; Leon 2014; Anzoategui et al. 2010; Claessens 2009). This approach measures the level of competition from the characteristics of industry. The n-firm concentration ratio (CR) and the Herfindahl–Hirschman Index (HHI) are usually used as competitive indicators. In empirical studies of traditional industry economics, these two indicators are the most widely used to assess competition. Calculating these specific indicators can reflect the importance of large firms at both inter- and intra-industry strata (Lipczynski et al. 2017).

The above two indicators have both strengths and weaknesses. Understanding the advantages and disadvantages can assist researchers in measuring and interpreting industrial competition in the right way. Measuring concentration is clear and simple because it is uncomplicated in the use of data (Leon 2014). However, in order to select suitable indicators, there are some important criteria that must be taken into account. Computing the CR may differ by the choice of the number of top *n* firms. This is because there are no rules on the choice. However, the choice of top *n* firms is not a critical point. The key issue for using the CR as indicator for measuring competition is the serious limitations on explaining the number and size distribution. It takes into account only the total value in the data of the top *n* firms (sales, assets, and employment), while the value outside those firms and the distribution within them are apparently ignored (Lipczynski et al. 2017). Regarding the Hannah and Kay's criteria, when there is a merger between incumbent firms, the concentration ratio should be increased; consequently, the competition level should be decreased. However, the outcome of the CR fails to satisfy this criterion if those incumbent firms are other small firms. Additionally, distributions within the top *n* firms are the vital factor. When there is a high skew distribution within the top *n* firms, the competition level should be lower than when there is a low skew or a symmetric distribution. Due to the limitations of the CR, the HHI is one of the most valuable indicators often used to measure market concentration. Compared with the CR, the HHI considers not only the number of all firms but also their size distribution (Lipczynski et al. 2017).

---

4　See Lipczynski et al. (2017, pp. 1–6).

To sum up, the CR can be used as a competition indicator if there is no change in the number of sellers and the size distribution does not vary. Otherwise, if there are structural changes in firms within the industry, such as mergers, acquisitions, new firms, and a highly skewed distribution of all firms in the industry, the HHI indicator can strongly represent market concentration.

*3.2. Nonstructural Approach*

The nonstructural approach is based on the new empirical industrial organization (NEIO)[5] and is more complicated than the structural approach in terms of both data requirements and measurement methodologies. Studies on the nonstructural approach do not infer that the market structure can identify the level of competition by indirectly observing a firm's conduct. It is possible to determine industrial competition from the firm's conduct instead (Lipczynski et al. 2017).

At this point, the area of empirical research in the NEIO has mainly focused on estimating behavioral equations. Empirical studies in the banking industry frequently employ balance sheets and income statements as proxies for output and input factors. Regarding the intermediate approach[6] to modeling in banking firms, total bank assets specified on the balance sheet statement are usually used as output. This is because they can explain not only loans producing but also other earning assets, such as securities or cash excess bank reserves. Three types of expenses are often used as proxies of input factors to produce bank assets; these are personal expenses, other nonfinancial expenses, and financial expenses, which represent labor, physical capital, and deposits, respectively (Leon 2014).

Even if there are many indicators under the nonstructural approach,[7] empirical studies on banking usually use the two main indicators, the Lerner Index (LI) and Panzar–Rosse H statistic (PRH) to measure competition (Kubo 2006; Anzoategui et al. 2010; Olivero et al. 2011; Fungáčová et al. 2013; Coccorese 2014; Rao Subramaniam et al. 2019). In practical terms, between these two indicators, choosing the proper one is important to interpret competition. Hence, the advantages and shortcomings should be clarified. The Lerner Index, one of the most useful and popular indicators, is used to measure a firm's market power from markup prices over marginal cost. One of the most distinctive features is the possibility to analyze the gradual evolution of individual bank pricing behavior over time. Moreover, the LI is flexible to observe the firm's market power in different market structures, since it does not require defining the market structure. However, under neoclassical theory, market power alone may not sufficiently explain the competition level. There are many factors that should be considered, such as product differentiation and entry–exit barriers. Moreover, computing LI by the conventional approach naturally assumes perfect technical and allocative efficiency, and it is difficult to demonstrate the circumstances of bank operation under perfect efficiency (Lipczynski et al. 2017; Leon 2014). In addition to the LI, the PRH has been widely applied to gauge rivals in the banking industrial. The PRH is one of the indicators under the static view of competition. The PRH always assumes the long-run equilibrium by applying the equilibrium of a monopolist, which is based on oligopoly theory. At the equilibrium, where marginal cost is equal to marginal revenue, when the bank's input price factors are raised, the bank's marginal cost will increase accordingly. The monopolist reacts to an increase of input price factors by a decrease in their quantity. Then, the bank's total revenues increase under the hypothesis that price elasticity of demand is greater than one. Panzar and Rosse (1987) showed that market competition can be measured by the sum of the elasticity of the firm's total revenues with respect to its factor input prices. The transmission from input factors to total revenues can reflect the degree of competition. The use of the PRH has an advantage in measuring competition across countries and less mature banking systems, which are often found in developing countries.

---

[5]　The NEIO comprises two generations: the first is based on oligopoly theory under the neoclassical concept, and the second is based on the Australian school concept (see Leon 2014).

[6]　Two approaches are used in modeling banking firms, production and intermediate (see Leon 2014).

[7]　The other indicators are rarely used in the banking industry. Some indicators, such as persistence of profit, are inappropriate for a developing country like Thailand (see Leon 2014).

Moreover, similar to the LI, the PRH does not require detailed specification of market definitions in order to estimate the revenue equations (Claessens and Laeven 2003; Sherrill 2004; Leon 2014).

Among these competition indicators, there is an unresolved conflict between theoretical and empirical evidence in previous studies on the impact of competition on financial stability and the effectiveness of monetary policy (Chileshe 2017). Some studies found that increased competition in the banking sector leads to strengthened stability in the financial system. This is because the low interest rates are likely to reduce payment defaults and systemic risk. As a result, the stability of the financial system is stronger (Boyd and Nicolo 2005; Tabak et al. 2012). At the same time, other views state that less competition in the banking sector leads to solid financial system stability. Due to the fact that less competition produces higher bank profitability, the bank's motivation to invest in high-risk assets is reduced, and subsequently, any crisis is more likely to be cushioned (Chileshe 2017; Tabak et al. 2012; Agoraki et al. 2011; Hellmann et al. 2000).

Several indicators are widely used to measure rivals in the competitive market of the banking sector. The main objective of the current study is to measure banking competition in Thailand by four indicators: concentration ratio (CR), Herfindahl–Hirschman Index (HHI), Lerner Index (LI), and Panzar–Rosse H-statistic (PRH). These indicators have different methods and factors. Previous studies argued that the indicators in the new empirical industry organization (NEIO) are the most useful, rather than the traditional industry indicators. Hence, this study focuses on calculating and measuring the individual indicators, and analyzing individual differences between them. Since approaches and indicators are options, understanding all dimensions is beneficial for further analysis of the impact of the degree of competition on banking market efficiency and policy effectiveness. Moreover, there are rarely studies on competition in the domestic banking industry. Therefore, this study provides an update on the level of competition of the banking industry in Thailand, and also clarifies the degree of competition in all aspects.

## 4. Data and Methodology

For the above reasons, the methodologies and calculations of banking competition indices in this study can be categorized into two groups: the structural approach, which includes the $CR_5$ and the HHI, and the nonstructural approach, which includes the LI and PRH.

### 4.1. Structural Approach

#### 4.1.1. Concentration Ratio

The $CR_5$ is calculated by the sum of asset shares of the five largest banks. The computation of the concentration ratio can be written as follows:

$$CR_n = \sum_{i=1}^{n} \frac{S_i}{S}; n = 1, \ldots, 5 \tag{1}$$

where $n$ is the number of banks; $S_i$ denotes the total assets of bank $i$; and $S$ denotes the total assets of all banks in the banking industry.

#### 4.1.2. Herfindahl–Hirschman Index

The HHI is calculated by the summing the square of asset shares of all banks in the banking system. The computation of HHI can be written as follows:

$$H = \sum_{i=1}^{N} \left[ \frac{S_i}{S} \right]^2 \tag{2}$$

where $S_i$ denotes the total assets of bank $i$, $N$ denotes the number of banks, and $S$ denotes the total assets of all banks in the banking industry.

### 4.1.3. Interpretation of Competition by CR$_5$ and HHI

The CR$_5$ takes into account size distribution of the top five banks. Its values can vary between 0 and 1. Greater values reflect a more highly concentrated market, which is likely to increase a bank's market power and contribute to low competition in the banking industry. In comparison to the CR$_5$, the HHI takes into account the size distribution of both the top five banks and banks outside the top five. According to the US Department of Justice and the Federal Trade Commission, the value of HHI can determine the degree of concentration on three strata: values lower than 1000 (or 0.1) determine low market concentration, values from 1000 to 1800 (or 0.1–0.8) determine moderate market concentration, and values higher than 1800 (or 0.18) determine high market concentration. The link between degrees of concentration and competition of HHI values is very similar to the CR$_5$, which implies that the higher the concentration, the less competitive behavior among firms.

### 4.2. Nonstructural Approach

#### 4.2.1. Lerner Index

Calculation of the LI considered in this study can be separated into two approaches, traditional and stochastic.

The computation of LI based on the traditional approach can be written as follows:

$$Lerner_{i,t} = \frac{P_{i,t} - MC_{i,t}}{P_{i,t}}; i = 1, \ldots, N \text{ and } t = 1, \ldots, T \tag{3}$$

where $P_{i,t}$ stands for the output prices set by bank $i$ at time $t$, and $MC_{i,t}$ stands for the marginal cost of bank $i$ at time $t$. Since the bank's marginal cost cannot be directly observed, it is necessary to identify the total cost function in order to obtain the bank's marginal cost.

Under the traditional approach, the calculation of LI is done with three steps: first, specify the total cost function of bank loan production; second, estimate the cost function and take the first derivative to obtain the bank's marginal cost; and third, use the marginal cost to calculate the LI according to Equation (3).

Based on the financial intermediaries approach, the bank's multiple output generally refers to total assets. The bank's total costs usually depend on only one output and three input prices. The three input prices widely used in the banking literature are the prices of labor, physical capital, and borrowed funds (Carbó et al. 2009; Beck et al. 2013; Fungáčová et al. 2013). Following these studies, the translog total cost function[8] can be specified as

$$\ln TC = \alpha_0 + \alpha_1 lnQ + \frac{1}{2}\alpha_2(lnQ)^2 + \sum_{j=1}^{3}\beta_j lnw_j + \sum_{j=1}^{3}\sum_{k=1}^{3}\beta_{jk} lnw_j lnw_k + \sum_{j=1}^{3}\gamma_j lnQlnw_j + \varepsilon \tag{4}$$

where the bank's output ($Q$) is total assets. The first bank's input price is the ratio of personal expenses to total assets ($W_1$), which represents the price of labor. The second is the ratio of non-interest expenses to fixed assets ($W_2$), which represents the price of physical capital. The third is interest expenses and short-term funding ($W_3$), which represents the price of borrowed funds. The cost function can be

---

[8]　This cost function is assumed to have symmetric and linear homogeneity restrictions in the three input prices.

estimated by the fixed effects method.[9] Then, the estimated coefficients from the cost function can be used to obtain the bank's marginal cost:

$$MC = \frac{TC}{Q}\left[\alpha_1 + \alpha_2 lnQ + \sum_{j=1}^{3} \gamma_j \ln w_j\right]. \tag{5}$$

The traditional Lerner Index in Equation (3) can be calculated once the marginal costs are made explicit.

The computation of the LI based on the stochastic approach can be written as follows:

$$L_{i,t} = \frac{\theta_{i,t}}{1 + \theta_{i,t}} \tag{6}$$

where $\theta$ is the estimated value of distance between technical efficiency and technical inefficiency in the production of banking firms, $i$ stands for an individual bank, and $t$ stands for time. Since the distances cannot directly observed, it is necessary to know the derivation of loan production in order to obtain the distance.

Based on the financial intermediaries approach, bank loans can be a single output of the bank's production, since credit activity is the most important part of banking business. Bank input prices can be classified into three factors, borrowed, physical capital, and labor. Following Coccorese (2014), the translog total cost function can be specified as

$$\begin{aligned} lnTC = {} & \alpha_0 + \alpha_1 lnQ + \sum_{h=1}^{3} \alpha_h lnW_h + \tfrac{1}{2}\alpha_{QQ}(lnQ)^2 + \tfrac{1}{2}\sum_{h=1}^{3}\sum_{k=1}^{3} \alpha_{hk} lnW_h lnW_k \\ & + \sum_{h=1}^{3} \alpha_{Qh} lnQ lnW_h + \alpha_E lnE + \tfrac{1}{2}\alpha_{EE}(lnE)^2 + \sum_{h=1}^{3} \alpha_{Eh} lnE lnW_h \\ & + \alpha_{EQ} lnE lnQ + \alpha_T T + \tfrac{1}{2}\alpha_{TT} T^2 \sum_{h=1}^{3} \alpha_{Th} T lnW_h + \alpha_{TQ} T lnQ \end{aligned} \tag{7}$$

where the bank's single output ($Q$) is the quantity of loans. The three bank input prices ($W_h$) are the ratios of interest expenses to total deposits ($W_{h=1}$), personal expenses to total assets ($W_{h=2}$), and other operating expenses to total fixed assets ($W_{h=3}$). The time trend ($T$) is included in order to explain technological change over the period. The bank's total equity is included in order to present the use of bank capital as a source of funds to invest in loan assets.

According to Coccorese (2014), banks produce loans by assuming profit maximization. Therefore, the possibility to produce loans depends on the condition that all banks will mark up their output prices not less than their marginal costs:

$$P_{it} \geq MC_{it}. \tag{8}$$

For the above equation, the maximum level of productive efficiency for the banking market can occur when output prices equal marginal cost. Otherwise, the production is inefficient. Hence, the greater the distance, the lower the productive efficiency.

From the empirical aspect, Equation (8) can be transformed[10] to revenue share to total costs (RC) and cost elasticity with respect to output ($E_{TC,Q}$), which can be written as follows:

$$RC_{i,t} \geq \frac{\partial \ln TC_{i,t}}{\partial \ln Q_{i,t}}. \tag{9}$$

---

9    See Appendix A.

10    The transformation is to multiply both terms of Equation (8) by output and total cost ratio $\left(\frac{Q}{TC}\right)$, to get $\frac{TR}{TC} \geq \frac{\partial \ln TC_{i,t}}{\partial \ln Q_{i,t}}$, where $\frac{TR}{TC} = RC$.

By taking the derivative of the total cost function in Equation (7), the cost elasticity with respect to output can be revealed:

$$\frac{\partial \ln TC_{i,t}}{\partial \ln Q_{i,t}} = \alpha_Q + \alpha_{QQ} ln Q_{i,t} + \sum_{h=1}^{2} \alpha_{Qh} \ln\left(\frac{W_{h,i,t}}{W_{3,i,t}}\right) + \alpha_{TQ}T + \alpha_{EQ} ln E_{i,t} + \varepsilon_{i,t}. \tag{10}$$

Since Equation (9) is under the condition of profit maximization, similar to Equation (8), it can be inferred that the distance between $RC_{i,t}$ (revenue to total cost ratio) and $E_{TC,Q}$ (cost elasticity with respect to output) demonstrates the bank's market power or productive efficiency for the banking market.

At this point, in doing empirical work, the stochastic cost frontier model[11] can be applied to estimate the technical inefficiency (TE) of bank production (Coccorese 2014). According to the stochastic cost frontier model, cost minimization is the maximum possibility of output that the bank can produce with given input factors. The stochastic cost frontier model of the bank's production can be written as

$$RC_{i,t} = \frac{\partial \ln TC_{i,t}}{\partial \ln Q_{i,t}} + v_{i,t} + u_{i,t} \tag{11}$$

where $i$ stands for an individual bank, $t$ stands for time, $RC$ or revenue–cost ratio is a dependent variable that stands for total production of a bank's output, $\frac{\partial \ln TC_{i,t}}{\partial \ln Q_{i,t}}$ are independent variables that stand for the deterministic part of the frontier, $v_{i,t}$ stands for the stochastic part of the frontier (a combination of these two is determined as the stochastic frontier), and $u$ stands for technical inefficiency (TE), which is a non-negative one-sided term.

For the above approach, following Coccorese (2014), an explicit model for estimating production and cost function can be rewritten as follows:

$$RC_{i,t} = \alpha_Q + \alpha_{QQ} ln Q_{i,t} + \sum_{h=1}^{2} \alpha_{Qh} \ln\left(\frac{W_{h,i,t}}{W_{3,i,t}}\right) + \alpha_{TQ}T + \alpha_{EQ} ln E_{i,t} + u_{i,t} + v_{i,t} \tag{12}$$

where the bank's output prices ($Q$) are defined as loans and other earning assets. The price of deposits ($W_1$) is defined as the ratio between interest expenses and total deposits. The price of labor ($W_2$) is defined as the ratio of personnel expenses to total assets. The price of capital ($W_3$) is defined as the ratio of other operating expenses to total fixed assets. The error term ($v$) is assumed to be independently normally distributed with zero mean and constant variance properties. Technical inefficiency $u$ is assumed to have half-normal distribution and non-negative value.

According to Coccorese's approach, Equation (12) can be estimated by maximum likelihood in order to obtain the distance between price and marginal cost ($\theta$). Then, the calculation of the Lerner Index can be written as follows:

$$L_{i,t} = \frac{\theta_{i,t}}{1 + \theta_{i,t}}. \tag{13}$$

To be more specific, there are four steps to calculate the LI by applying the stochastic cost frontier model: (1) estimate the production and cost function with Equation (13), (2) predict the technical inefficiency ($u_{i,t}$) with Equation (12), (3) predict the revenue–cost ratio ($RC_{i,t}$) with Equation (12), and (4) calculate the LI according to Equation (13).

### 4.2.2. PRH

Calculating PRH in an empirical work can be separated into two steps. The first step is to estimate the dynamic revenue equation for bank-level data. This study follows the methodology of Goddard and John and Wilson (2009) and Olivero et al. (2011), and the estimating model can be written as

---

[11] See Appendix A.

$$\ln(R_{i,t}) = \beta_1 \ln(R_{i,t-1}) + \beta_2 \ln(W_{1\ i,t}) + \beta_3 \ln(W_{2\ i,t}) + \beta_4 \ln(W_{3\ t,t}) + x'_{i,t}\gamma + e_{i,t} \tag{14}$$

where $i$ stands for an individual bank; $t$ stands for time; $R$ stands for the bank's total revenues; ($W_1$) stands for input factor prices of the bank's deposits, which is the ratio of interest expenses to total assets; ($W_2$) stands for the input factor prices of the bank's capital, which is the ratio of non-interest expenses to total assets; ($W_3$) stands for the input factor price of labor, which is the ratio of personnel expenses to total assets; $x$ stands for the vector of control variables, which are the ratios of equity to total assets, loans to total assets, and other revenues to total assets; and $e$ is a random disturbance term. It has been criticized in previous studies that an analysis of PRH in a static equation would be biased toward zero. This is because the bank market is always assumed in long-run equilibrium each time. Therefore, Arellano and Bond's (1991) generalized method of moments (GMM)[12] is employed to estimate Equation (13) (John and Wilson 2009; Olivero et al. 2011).

After estimating Equation (14), the second step is to calculate the sum of long-run elasticity of the bank's total revenue with respect to each of its factor input prices. The sum of elasticity is the H statistic, which was introduced by Panzar and Rosse (1987). The formula can be written as follows:

$$\frac{\beta_1 + \beta_2 + \beta_3}{1 - \beta_0}. \tag{15}$$

### 4.2.3. Interpretation of Competition by LI and PRH

The LI identifies the market power of banks by investigating the ratio of difference between price and marginal cost. This can reflect the competition behavior among banks. The bank's market power is assumed to have a negative relationship with competitive behavior. More market power leads to less competitive behavior in the banking industry. The LI has a range between 0 and 1. A value close to zero reflects a highly competitive market. A number of recent studies have pointed out that the Lerner Index is a beneficial index that thoroughly measures individual bank-level behavior.

PRH has different values, which leads to different interpretations of bank conduct. The interpretation is specified in Tables 1 and 2.

**Table 1.** Classifying market power of industries with the Lerner Index (LI).

| Market Power | Value |
| --- | --- |
| Monopoly | $L = 1$ |
| Perfect Competition | $L = 0$ |

Source: (Lipczynski et al. 2017, p. 74).

**Table 2.** Classifying industries with Panzar–Rosse H statistic (PRH).

| Market Structure | Value of PRH |
| --- | --- |
| Monopoly | $H \leq 0$ |
| Monopolistic Competition | $0 < H < 1$ |
| Perfect Competition | $H = 1$ |

Source: (Lipczynski et al. 2017, p. 374).

### 4.3. Data Descriptions

This study uses quarterly panel data on bank balance sheets and income statements from Q1 2001 to Q1 2019, which is the period after the end of the International Monetary Fund (IMF) bailout program. The data are available online from the Securities and Exchange Commission (SEC) of Thailand. Other financial institutions, such as financial companies, securities companies, foreign bank branches, and

---

12　See Appendix A.

state enterprise banks, are excluded from the study because they are dissimilar in business scope, capital structure, and regulatory environment. The sample is an unbalanced panel totaling 795 observations. The descriptive statistics of the variables used in the empirical specification are reported in Table 3.

**Table 3.** Descriptive statistics of variables used to calculate the indices.

| Variable | Description | Mean | SD | Min | Max |
|---|---|---|---|---|---|
| Total revenue (1) | Sum of interest income and non-interest income | 13,452.35 | 12,271.11 | 27.58752 | 47,932.95 |
| Total cost (1) | Sum of interest expenses, personal expenses, and other operating expenses | 8756.12 | 7257.953 | 23.24 | 27,853.04 |
| Revenue share to total cost (2) | Total revenue to total cost ratio | 1.47 | 0.31 | 0.4 | 2.88 |
| Bank output (1) | Sum of loans and other earning assets | 885,558.60 | 813,554.80 | 8360.36 | 3,106,581 |
| Price of deposits (2) | Interest expenses to total deposits ratio | 7.63 | 4.73 | 2.06 | 47.64 |
| Price of labor (2) | Personal expenses to total assets ratio | 0.002 | 0.001 | 0.000 | 0.035 |
| Price of capital (2) | Other operating expenses to total fixed assets ratio | 0.1 | 0.08 | −0.11 | 0.95 |
| Total equity (1) | Sum of bank's equity | 87,645.94 | 92,685.39 | 109.44 | 413,378.9 |
| Total assets (1) | Sum of bank's total assets | 900,147.70 | 819,463.20 | 308.48 | 3,071,110 |
| Equity to total assets (2) | Equity to total assets ratio | 0.01 | 0.04 | 0.01 | 0.35 |
| Loans to total assets (2) | Loans to total assets ratio | 0.71 | 0.11 | 0.19 | 1.01 |
| Other operating income to total assets (2) | Other operating income to total assets ratio | 0.00 | 0.00 | −0.01 | 0.04 |

(1) Constant million THB; (2) ratio, 796 observations.

## 5. Empirical Results

As outlined in the introduction, it has been not entirely clear which competition indicators are suitable for the Thai banking industry. This section begins to analyze outcomes of the structural and nonstructural approach. The explanations of both approaches are based on an analysis of trends and the value of each indicator in order to interpret competitive behaviors and identify the best measurement, and the implications of proceeding with policies for banks.

### 5.1. Structural Approach

Figure 1 presents trends of the indicators $CR_5$ and HHI during Q1 2001 to Q1 2019. The $CR_5$ has an upward trend, which means that the Thai banking industry tends to be more concentrated and less competitive over time. In contrast to the $CR_5$, the HHI shows a downward trend, which means that the Thai banking industry tends to be less concentrated and more competitive over time.

Average values of the $CR_5$ and HHI are presented in Table 4. The $CR_5$ has an average value of 0.91 and a range of 0.83 to 0.95. The HHI has an average value of 0.15 and a range of 0.14 to 0.18. In comparison, the $CR_5$ shows that the Thai banking industry is nearly noncompetitive, whereas the HHI reveals moderate competition. Interestingly, it is obvious that even though both the $CR_5$ and the HHI are under the same approach, the results present the tendency of Thai banking competition in different ways.

**Table 4.** Outcomes of concentration ratio ($CR_5$) and Herfindahl–Hirschman Index (HHI).

| Indicator | Mean | SD | Min | Max |
|---|---|---|---|---|
| $CR_5$ | 0.91 | 0.04 | 0.83 | 0.95 |
| HHI | 0.15 | 0.01 | 0.14 | 0.18 |

Author's calculation.

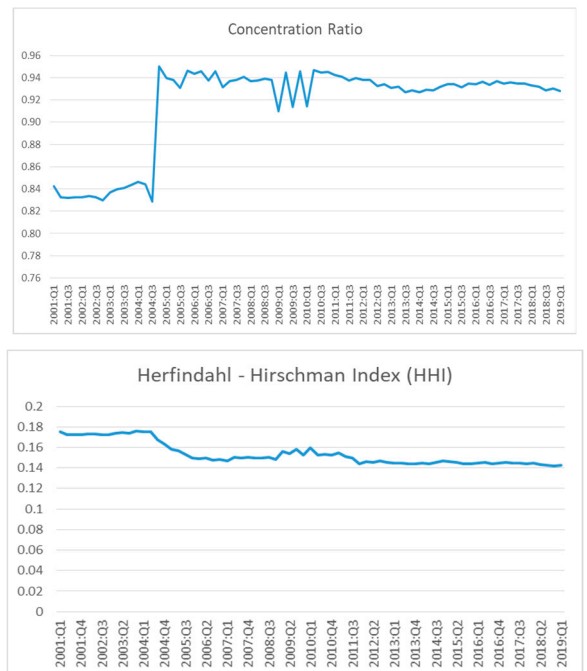

**Figure 1.** Trends of bank competition under the structural approach. Source: Authors' calculation.

At this point, with regard to the different results of the $CR_5$ and HHI, it is significant to discuss why they assess competition in different ways and which indicator can accurately measure a fit state of competition in the Thai banking sector. Based on the structural approach, the explanation of different outcomes can be divided into two reasons. The first is derived from the formulation for calculating of these indicators. The calculation of the $CR_5$ considers the asset shares of the five largest banks only. Other banks seem to lack consideration. Thus, if there has a change in bank assets outside the largest five banks, the $CR_5$ would fail to explain the characters of banking market structure accurately. Meanwhile, the HHI considers the asset shares and size distribution of all banks in the banking system. Hence, it can capture all structural changes within the industry. The second is the facts that Thai banking industry experienced changes in size distribution many times during the period of this study. Besides, the changes originated from outside the top five banks. The Thai banking industry crucially changed from the significant circumstances of mergers and acquisitions activities, such as Bank of Asia and UOB Ratanasin bank merging with United Overseas Bank, and CIMB acquiring Bank Thai. Additionally, medium-size banks, including Bank of Ayudhya (BAY), Thai Military Bank (TMB), and Thanachart bank (TBANK), accepted a share acquisition in order to target the strengths and capture market synergies. Moreover, three new banks entered the industry during 2005–2007, TISCO, LH, and TCR respectively. This reflects a change in the size distribution of the banking sector, which is a key consideration for using the concentration measure. Obviously, these changes occurred sporadically outside the five largest banks. According to the above reasons, we may not conclude that HHI is the best measurement of the Thai banking competition under the structural approach. This is because the theory underlying of this approach, which is the SPC paradigm has been heavily criticized. It has been attacked by the contestable theory, which states that even if market structure is characterized as highly concentrated, it can behave competitively if obstacles to entry and exit are low. The SPC paradigm states that market structure can determine firm's behavior then the behavior can determine profitability. If banking market characterized as a high concentration, the level of collusion will increase, contributing to high profitability, which implies lower competition. In case of Thailand, the outcome of $CR_5$ and HHI presents a very high level of concentration in banking market. Furthermore, the results is remarkably similar to actual data of market shares, which proxy by total assets in Thai banking

sector. As can be seen in Figure 2, the market shares of top five banks are BBL18%, SCB 17%, KBANK and KTB 16%, and BAY 12%.

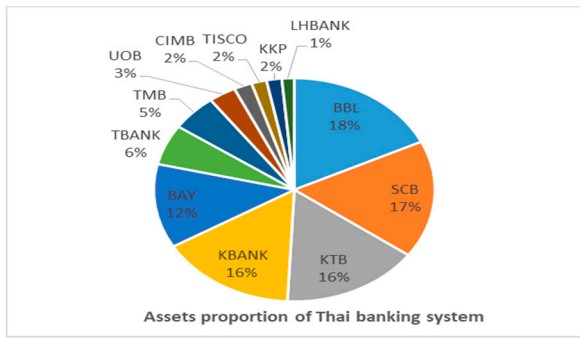

**Figure 2.** Market share of Thai banking sector during 2001–2009. Source: the security and exchange commission, Thailand (SEC). (www.sec.or.th 2020).

It can be conclude that market structure of the banking sector in Thailand is considerably concentrated therefore it possibly contribute high bank's profitability according to the SPC paradigm. However, when we consider the Net Interest Margin: NIM (see Figure 3), which is a proxy for reflecting bank's profitability. It is obvious that the bank profitability tends to stable which not represents high profitability and also contradicts to the SPC theory. Moreover, this disaggregate data of the NIM indicates that each bank is less likely to set its interest rates different from other banks. The more concentrated in Thai banking sector might not determine bank's market power. Therefore, competition level in Thai banking sector cannot identify based on the structural approach. For these reasons, in view of the structural approach, this study concludes that the indicators, which measure from the structural approach are not suitable for interpretation of competitive environment in the Thai banking sector.

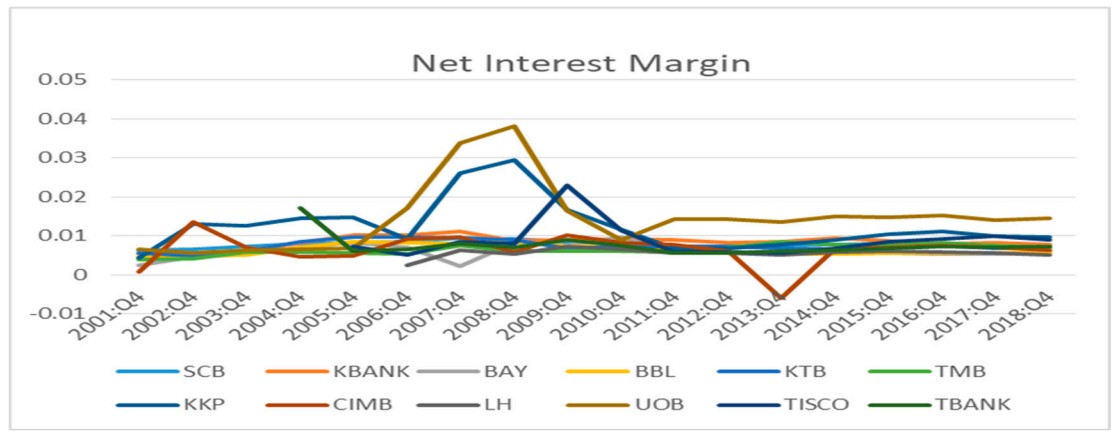

**Figure 3.** Net interest margin disaggregated by individual banks over the period 2001–2018. Source: Authors' calculation.

### 5.2. Nonstructural Approach

Under the nonstructural approach, the indicators LI and PRH are used as empirical measures of competition in the Thai banking industry. Assessing competition by the PRH has limitations for trend analysis, because it can observe only one outcome of competition on the long-run equilibrium. However, it is tremendously useful for cross-country studies, as widely applied in much of the literature. With regard to average value analysis, the PRH (see Table 5) has an average value of 0.17 over the period of study (Q1 2001 to Q1 2019). It can be inferred that the Thai banking industry has moderate competitive behavior among banks.

Table 5. Outcomes of PRH and LI. SFA, stochastic frontier analysis.

| Approach | Indicator | Mean | SD | Min | Max |
|---|---|---|---|---|---|
| | Panzar–Rosse H statistic | 0.17 | 0.03 | - | - |
| Nonstructural | Lerner Index (SFA method) | 0.40 | 0.03 | 0.36 | 0.60 |
| | Lerner Index (traditional method) | 0.37 | 0.17 | −1.25 | 0.69 |

Author's calculation.

In addition to the PRH, calculation by the LI in this study is divided into two methods, traditional and stochastic frontier. A trend analysis is illustrated in Figure 4. Regarding the SFA method, the average LI presents a downward trend, which means a decline of market power and a tendency toward more competition over time. On the other hand, the traditional LI shows an upward trend, though somewhat fluctuating, which indicates a rise in bank market power and a decrease in bank competition. The average value of the LI (see Table 5) obtained from the SFA method is 0.40, and it varies from 0.36 to 0.60. The average value using the traditional method is relatively close to that of the SFA method at 0.37, while the range is quite wide, −1.25 to 0.69. The average values of the LI calculated by the two methods are quite similar, while the ranges are very different. The range values of LI (SFA) is more reliable than LI (FE), since the theoretical concept states the range value between zero to one. Zero value defined as perfect competition and value of one defined as pure monopoly. Outside the range of zero to one cannot be defined about competition level. In addition, the range of values, which below to zero or exceed unity can be indicated the market power as well. However this study only specifies the best measurement for evaluating the level of competition in Thai banking sector. Moreover, the average values of LI (SFA) equal to 0.4 is adjacent to zero which implies as competitive in the sector. As a result, we believe that LI(SFA) is greater explained the study of competition in Thai banking sector.

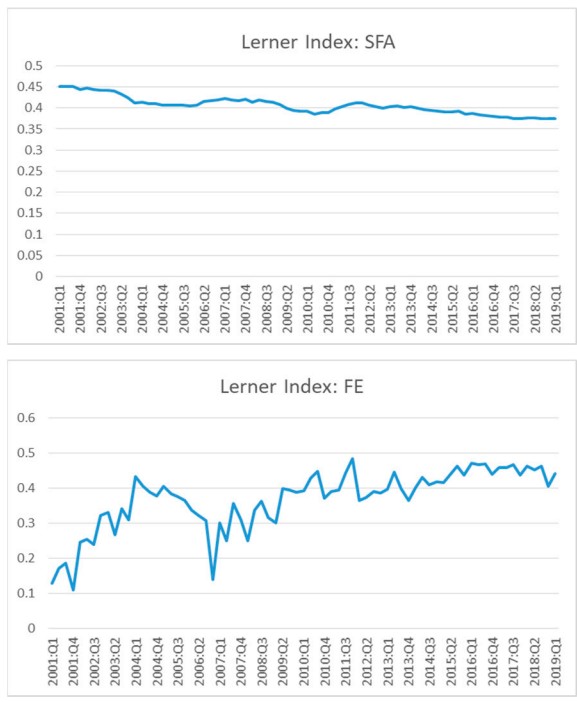

**Figure 4.** Trends of bank competition under the nonstructural approach. Source: Authors' calculation.

Calculating the three indicators produces different outcomes, which causes a problem in interpreting the competitive environment. Therefore, this study attempts to identify the best indicator that can accurately measure competition in the Thai banking industry. For the result of the PRH, since this indicator is valid in the long–run equilibrium condition, which is hard to achieve. It has

been argued that using the H statistic can result in bias (Claessens and Laeven 2003; Shaffer 1983). Bikker et al. (2006) claim that there is misspecification of the calculation of the H statistic. They found that the use the ratio of total income to total asset as the endogenous variable when calculating the H statistic will lead to an overestimation of the competition degree. In addition, Claessens and Laeven (2003) point out that this statistics tends to be biased when the bank sample size is very small (below 20 banks). At this point, because of the small sample size of the Thai banking sector (12 banks), the LI is appropriate as a reliable indicator to represent competition.

Regarding the outcomes obtained from empirical measurement of the LI, the calculation by the SFA method is considered to be a suitable technique for the Thai banking industry. The reasons can be divided into two aspects, the first aspects is all the non-negative value of the LI (SFA). In an explanation of competition, the negative values can explain the bank's market power but cannot interpret competition level. Some studies point out that the negative value of LI can occur when markup price is lower than marginal cost in the short run. However, no conclusions can be drawn about competitive environment under the negative value of LI (FE). The second aspect is correlation between the LI, calculating from SFA method, and banking industry-specification of Thailand. Banking industry-specification can be divide into five variables (see Ghosh 2018). Bank profitability is represented by return of assets (ROA), which is defined as bank's net income after-tax to average total assets. It is expected a positive relationship between LI (SFA) and bank's profitability. The higher competition stems from the lower profitability ratio. This matters to lower value of LI (SFA). Diversification is defined by share of non-interest income to total income. It is expected a negative relationship between LI (SFA) and bank's diversification. The higher competition is caused by the higher non-interest incomes to total incomes ratio. Due to the facts that banks can expand their variety of financial services which is fostering competitive environment. This matters to the lower value of LI (SFA). Cost efficiency is represented by bank cost to income ratio, which is defined as the share of bank's operating expenses to banks total revenues. It is expected a negative relationship between LI (SFA) and the cost efficiency. The higher competition is a consequence of the less effective costs (a high value of cost-to-income ratio), leading to higher marginal cost and lower profits. This matter refers to the lower value of LI (SFA). Capitalization is represented by total equity capital to assets ratio. It is expected a positive relationship between LI and the capitalization. The higher competitive environment is due to lower capitalization, which contributes to lower the bank's market power. This matter refers to the lower value of LI (SFA). As can be seen in Appendix B Table A1, Correlation between LI (SFA) and bank's profitability, capitalization, diversification variables are statistically significant at 1%. In addition, correlation between LI (SFA) and bank's cost efficiency is statistically significant at 5%. All these variables have relationship as expected, except for capitalization. This reflects to the fact that the high bank capital structure may not leads to the high market power in the Thai banking industry. Additionally, as the Thai economy depends heavily from banking sector, the higher competition should be caused the economic growth. Thus a negative relationship is hypothesized between GDP and the LI (SFA). The result reports that correlation between the LI (SFA) and GDP is positive and significant at level 1%. Up to this point, this study concludes that using competition measurement by the structural approach, which are $CR_5$ and HHI indicator may ineffective in the context of Thai banking sectors. The non-structural approach, which is LI calculated by the SFA method is a good indicator of interpretation of Thai banking industry.

## 6. Conclusions

Thailand is one of developing countries, which the banking industry plays a significant role in the economy because banks hold a large amount of financial assets and their business activities are related to economic agents, which are household savings and business sector investments. At this point, this sector is regulated by policy-makers, therefore a change in its competitive environment could significantly affect not only the economy but also policy effectiveness.

For the above reason, this paper was aimed at computing the competition indices in the Thai banking sector during the period from Q1 2001 to Q1 2019 by using bank-level data. We focused on the

four indicators of competition that are usually found in the banking literature: the concentration ratio of the five largest banks (CR$_5$), the Herfindahl–Hirschman Index (HHI), the Lerner Index (LI), and Panzar–Rosse H statistic (PRH). Several methods were employed to gauge the degree of competition through these specific indicators. Since the indicators have different factors and methods, their outcomes may be inconsistent. Furthermore, numerous studies have used one indicator over another, although there is no consensus on which one is better for the Thai banking sector. The findings of this study indicate that measuring competition based on the structural approach is inappropriate in the context of Thai economy. One of the key reason is the result from empirical measurement incompatible with the SPC paradigm. Thai banking sectors characterize as a high concentrated banking system while the net interest margin of banks are not difference and the number of banks are limited. The high concentration may not lead to high profitability behavior of banks. Therefore, to measure bank competition by using this approach may causes to misleading.

With regard to the nonstructural approach, we found that the Lerner Index calculated by the stochastic frontier method represents a useful indicator. This is because all values of Lerner Index from this method reveal a non-negative value, which can explain bank competition by implying from the market power values. Besides, the inference of the competitive environment in the banking industry is consistent with the true circumstances and character of Thai banking system.

The role of competition in the banking sector has some special properties that differ from other industries. Competition among banks is important for the efficiency of the industry, similar to other industries. At the same time, the stability of the banking system is also crucial for effective supervision. The trend in the Thai banking industry today is moving toward digital banking. This digital transformation is leading to a change in the competitive environment. Policy-makers should consider the effect of the change when controlling and regulating monetary policy. If they do not carefully control the competitive environment in the country, it will probably raise inefficiency or instability in the economy. This paper provides a better understanding of competition indicators for the Thai banking sector. It will useful for further study in order to test the efficiency of this industry or the effectiveness of monetary policy through this sector in Thailand by using the reliable indicators shown in this study. However, there are some limitations in this paper: in addition to market power and industrial concentration, the level of competition can be measured in different dimensions, such as product differentiation, barriers to entry and exit, numbers of buyers and sellers, level of technological progress, level of access to banking services, and level of information. Thus, further research could extend the investigation in this area by focusing on other approaches.

**Author Contributions:** Conceptualization, J.P., T.L. and B.C.; Methodology, J.P., T.L. and B.C.; Software, J.P. and B.C.; Validation, T.L. and B.C.; Formal Analysis, J.P. and T.L.; Data Curation, J.P.; Writing—Original Draft Preparation, J.P.; Writing—Review & Editing, T.L.; Visualization, J.P.; Supervision, T.L. and B.C. All authors have read and agreed to the published version of the manuscript.

**Funding:** This research received no external funding.

**Conflicts of Interest:** The authors declare no conflict of interest.

## Appendix A

*Appendix A.1. Stochastic Frontier Model*

Stochastic frontier analysis (SFA) is usefully employed to calculate the Lerner Index in Section 4. The stochastic frontier model can capture distance, which represents technical inefficiency in the production function in the banking industry.

The model specification can be written as

$$lny_{it} = \alpha + x_{it}'\beta + v_{it} + u_{it} \quad i = 1, 2, \ldots, N, \text{ and } t = 1, 2, \ldots, T \tag{A1}$$

where $i$ represents a bank and $t$ represents time, $y_{it}$ is a dependent variable, $\alpha$ is an intercept term, $x_{it}$ stands for the $(k \times 1)$ vector of independent variables, $\beta$ stands for the $(k \times 1)$ vector of coefficients, $v_{it}$ is a disturbance term, and $u_{it}$ stands for technical inefficiency (TE).

Technical inefficiency, $u_{it}$, should have a non-negative value in this analysis because the key consideration in this study is cost minimization. In general, firms should be setting their output prices over their costs. In a perfect competitive market, the setup prices of firms should be equal to the marginal cost of production, which reflects the lowest possible value of the firm. The cost minimization frontier problem represents the lowest boundary of a firm in any produced output. There is no possibility to set prices lower than the marginal cost in a competitive market because it will cause firms to leave the market. In other words, the frontier illustrates the technical efficiency of setting up prices in the market. However, in general situations, prices can be marked up higher than marginal costs due to higher market power of firms, differentiated products, and asymmetric information of buyers and sellers. This circumstance will reflect the technical inefficiency of the market, which is represented by output prices higher than efficiency cost at any output level. Hence, the distance between markup prices and the lowest boundary of the frontier will show the non-negative value of technical inefficiency.

However, Equation (A1) cannot be directly estimated by ordinary least squares (OLS) because there are two error components. Estimating by OLS will lead to inconsistent estimators and unpredictable technical efficiency. The maximum likelihood (ML) method is more widely used in this modelling. Some additional assumptions are required for the two error components, as follows:

1.　The probability density function (PDF) of the disturbance term has symmetric distribution.
2.　The two component errors, $v_{it}$ and $u_{it}$, are statistically independent of each other.
3.　The two component errors, $v_{it}$ and $u_{it}$, are independent and distributed across observations.

Therefore, based on the above assumptions, it is necessary to form a joint density function of $v_{it}$ and $u_{it}$, which can be written as

$$f_{v,u}(v_{it}, u_{it}) = f_v(v_{it}) f_u(u_{it}) \tag{A2}$$

where $f_v(v_{it})$ is the PDF of the error term and $f_u(u_{it})$ is the PDF of the technical inefficiency. Then, Equation (A2) can be formulated as

$$f_{\varepsilon,u}(\varepsilon_{it}, u_{it}) = f_v(\varepsilon_{it} - u_{it}) f_u(u_{it}) \tag{A3}$$

where $\varepsilon_{it} = v_{it} + u_{it}$. Note that the Jacobian transformation from $(v_{it}, u_{it})$ to $(\varepsilon_{it}, u_{it})$ is equal to 1. To find the probability density function of $\varepsilon_{it}$, the values of $u_{it}$ need to be integrated out of Equation (A3). Then we get the marginal PDF of $\varepsilon_{it}$ as

$$f_\varepsilon(\varepsilon_{it}) = \int_0^\infty f_u(u_{it}) f_v(\varepsilon_{it} - u_{it}) du_{it}. \tag{A4}$$

From Equation (A4), the log likelihood function is as follows:

$$lnL\left(\alpha, \beta, \sigma_u^2, \sigma_v^2 \middle| lny_{it}, x_{it}\right) = lnf_\varepsilon\left(y_{it} - \alpha - x_{it}'\beta \middle| \sigma_u^2, \sigma_v^2\right). \tag{A5}$$

From Equation (A5), estimation by ML can obtain consistent estimators and predict the technical inefficiency. In this study, half-normal distribution is assumed for the technical inefficiency.

*Appendix A.2. Fixed Effects Model*

According to the measure of banking competition through the traditional Lerner Index in Section 4, the fixed effects model can be used to estimate the total cost function (Fungáčová et al. 2013). The model specification can be written as

$$y_{it} = x_{it}'\beta + c_i + u_{it} \quad i = 1, 2, \ldots, N \text{ and } t = 1, 2, \ldots, T \tag{A6}$$

where $i$ represents a bank and $t$ represents time, $y_{it}$ is a dependent variable, $x_{it}$ stands for the $(k \times 1)$ vector of independent variables, $\beta$ stands for the $(k \times 1)$ vector of coefficients, $c_i$ stands for the time-invariant unobserved effect, and $u_{it}$ is a disturbance term, which is assumed to be identically and independently distributed (i.i.d.).

Under the fixed effect approach, $c_i$ is assumed to be correlated with the independent variable $(x_{it})$, which causes an endogeneity problem. In general, an instrumental variable is a useful tool in order to solve the problem. However, instrumental variables cannot apply in this case. This is because $c_i$ is unobservable.

Since $c_i$ cannot be estimated, the model should be transformed to eliminate the unobserved effect variables by taking mean difference[13] with Equation (A6), which can be written as

$$y_{it} - y_i = (x_{it} - x_i)'\beta + (u_{it} - u_i), \quad i = 1, 2, \ldots, N \text{ and } t = 1, 2, \ldots, T \tag{A7}$$

where $y_i$ is $\frac{1}{T}\sum_{t=1}^{T} y_{it}$, $x_i$ is $\frac{1}{T}\sum_{t=1}^{T} x_{it}$, and $u_i$ is $\frac{1}{T}\sum_{t=1}^{T} u_{it}$. Then, Equation (A7) can be estimated by OLS; the estimated parameters are unbiased and consistent properties (Wooldridge 2010).

*Appendix A.3. Dynamic Panel Model*

According to the measure of bank competition by the PRH in Section 4, the dynamic panel model is employed to estimate the elasticity of total revenue with respect to input factors. Following Olivero et al. (2011), the model specification can be written as

$$y_{it} = \rho y_{it-1} + x_{it}'\beta + c_i + u_{it}, \quad i = 1, 2, \ldots, N \text{ and } t = 1, 2, \ldots, T \tag{A8}$$

where $i$ represents a bank and $t$ represents time, $y_{it}$ is a dependent variable, $y_{it-1}$ is a lagged dependent variable, $\rho$ stands for coefficients corresponding to a lagged dependent variable, $x_{it}$ stands for the $(k \times 1)$ vector of independent variables, $\beta$ stands for the $(k \times 1)$ vector of coefficients, $c_i$ stands for the time-invariant unobserved effect, and $u_{it}$ is a disturbance term, which is assumed to be iid. The assumption of the unobserved effect variable $c_i$ is similar to Equation (A6). Therefore, by taking the first difference of Equation (A8), the model can be written as follows:

$$y_{it} = \rho y_{it-1} + x_{it}'\beta + u_{it}, \quad i = 1, 2, \ldots, N \text{ and } t = 1, 2, \ldots, T \tag{A9}$$

where $y_{it}$ stands for $(y_{it} - y_{it-1})$, $y_{it-1}$ stands for $(y_{it-1} - y_{it-2})$, $x_{it}$ stands for $(x_{it} - x_{it-1})$, and $u_{it}$ stands for $(u_{it} - u_{it-1})$. However, there is a correlation between the explanatory variable $(y_{it-1})$ and the error term $(u_{it})$, which produces an inconsistent estimated coefficient. Although sequential exogeneity is assumed, the problem still occurs: $E(y_{it-1}, u_{it}) \neq 0$.

In general, Anderson and Hsiao (1981) suggested that in order to obtain efficient estimators, the valid instrument variable should be $y_{it-2}$, which not only is uncorrelated with $(u_{it} - u_{it-1})$ but also has high correlation with $y_{it-1}$ Besides, to gain more efficient estimators, Arellano and Bond (1991) indicated that adding instrumental variables in the first differenced equation is the best way to

---

[13] Note that there are alternative ways to transform the model by taking the first difference as well.

improve model efficiency. The available instrumental variable, as suggested by Arellano and Bond (1991), should be $Z = (y_{it-2}, y_{it-3}, \dots, y_{i1})$. However, using many instrumental variables will cause an overidentification problem. Equation (A9) can be estimated by generalized method of moments (GMM). The estimated parameters will be consistent (Cameron and Trivedi 2005).

## Appendix B

**Table A1.** Correlation between LI (SFA) and banking industry-specification.

| Variables | Correlation | *p*-Value |
|---|---|---|
| GDP | −0.5415 | 0.0000 |
| Profitability | 0.1914 | 0.0000 |
| Capitalization | 0.6591 | 0.0000 |
| Diversification | −0.3921 | 0.0000 |
| Cost efficiency | −0.0852 | 0.0166 |

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
