# Peer review of "Empirical Measurement of Competition in the Thai Banking Industry"

_economies, doi:10.3390/economies8020044_

Round 1

Reviewer 1 Report

The paper expends a great deal of computational effort to address a relatively simple question--the degree of competitiveness of the Thai banking sector. However, it lacks a well-defined research question or way of answering it. Since there are only 12 banks, and the top 5 banks have a market share of 93%, it seems pretty clear that the sector is rather concentrated. It would have been useful to provide the actual data on market shares in the sector.

The methodology is fine, but the discussion can be improved significantly. The discussion of the results for the CR5 and HHI indices concludes that the values move in opposite directions and have opposite implications, but this was due to a one-off event in 2004, the reason for which should be explained. Since then both indices are roughly flat, and the difference in interpretation is probably minimal. The discussion of the difference between LI and PRH is moot, since the latter only applies to long-run equilibrium and needs at least 20 firms. The implications of the LI computation are not clear, since the trends of the SFA and FE are in the opposite direction, and it is merely concluded that the values are "in the theoretical range." So it is not clear what the point of the exercise is.

The paper needs a more interesting research question and method of resolving it.

Minor points:

Figure 3 is not very readable

There are extensive equations in the text, but the more complicated ones can be relegated to the appendix, since they are not original.

Author Response

Dear Sir,

Thank you very much for your comments. Please see the revision of manuscript in the attached file. Your comments listed below in bold. I have revised the manuscript according to such comments as follows:

The paper expends a great deal of computational effort to address a relatively simple question--the degree of competitiveness of the Thai banking sector. However, it lacks a well-defined research question or way of answering it. Since there are only 12 banks, and the top 5 banks have a market share of 93%, it seems pretty clear that the sector is rather concentrated. It would have been useful to provide the actual data on market shares in the sector.

Response: I tried to improve the way of answering the research question, which indicates in the next below response. 

The methodology is fine, but the discussion can be improved significantly. The discussion of the results for the CR5 and HHI indices concludes that the values move in opposite directions and have opposite implications, but this was due to a one-off event in 2004, the reason for which should be explained. Since then both indices are roughly flat, and the difference in interpretation is probably minimal. The discussion of the difference between LI and PRH is moot, since the latter only applies to long-run equilibrium and needs at least 20 firms. The implications of the LI computation are not clear, since the trends of the SFA and FE are in the opposite direction, and it is merely concluded that the values are "in the theoretical range." So it is not clear what the point of the exercise is.

Response: I am really sorry that makes you confuse. I added more explanation and changed some interpretation of the result for the CR5 and HHI. Please see the revised between line no. 439 – 456. For the result of PRH, I have already clarified the unclear sentence. Please see the revised between line no. 479 – 487.For the results of LI (FE) and LI (SFA), I have revised and increased the more empirical evidences by using the unique data set of Thai banking system, which can improved the better understanding of the sector.  Please see the revised between line no. 479 – 552. For the word of “in the theoretical rang”, I rewritten by incremental more explaining. Please see the more explaining between line no. 490 – 494.

The paper needs a more interesting research question and method of resolving it.

Response: Thank you very much for your suggestion. I will keep improving our work for the further paper.

Best Regards,

Jirawan P.

Reviewer 2 Report

Journal: Economies (ISSN 2227-7099)

Manuscript ID: economies-740356

Manuscript Type: Article

Title: Empirical Measurement of Competition in the Thai Banking Industry

Review Report

This study investigates the following: 1) which proxy is a better indicator among four indicators, namely concentration ratio, Herfindahl-Hirschman Index, Lerner Index, and Panzar-Rosse H statistic, and 2) how to measure the competitiveness of the Thai banking industry. It finds that 1) the Lerner Index, measured by stochastic frontier analysis, is the most reliable indicator of the banking competition environment within Thailand, and 2) for the period from Q1 2001 to Q1 2019 the degree of Thailand banking competition have increased – which is in accordance with the Financial Sector Master Plan of Thailand government. Based on their empirical findings, implying the increase in the banking sector competition, the authors suggest that policymakers should carefully regulate competition policy by considering the systematic risk of the banking system. In sum, I believe this study asks interesting, relevant research questions which can provide some additional evidences for readers to understand Thai Banking industry. I think that a couple of changes/modifications would help authors build a stronger case. With your permission, I would like to share a couple of suggestions. That said, I will go straight to the points:

Why Competition? Competition is one of very many factors which would affect the effectiveness and/or efficiency of banking industry – yet, it is clearly not the only one. For example, monitoring mechanism may possibly replace the role that competition provides to the market. What about the changes in government regulations and/or foreign direct investments (FDI)? Then, it is my recommendation to provide more explanation/justification why one should focus solely on competition to understand the Thailand banking industry.

Providing incremental contribution to the existing literature: I think that one of the strengths of this study is dealing with the unique dataset from Thailand. I hope that it provides some empirical evidence from the collected Thai dataset and more importantly provide additional explanation which can be used in understanding the Thailand Banking industry beyond and above the earlier findings from other countries. I think that it should be considered as the main point of the study.

Writing: I think that current manuscript can be benefited if it can be edited by a professional English-native writer. I saw a few disconnections in writing (between paragraphs and sentences). Please consider proofreading before resubmitting your manuscript.   

In sum, I again thank you for giving this opportunity to learn from your research project and I wish you the very best.

Best Regards,

Author Response

Dear Sir, 

Thank you very much for your comments. Please see the revision of manuscript in the attached file. Your comments listed below in bold. I have revised the manuscript according to such comments as follows:

The reviewer recommends to provide more explanation/justification why one should focus solely on competition to understand the Thailand banking industry as well.

Response: Thank you very much for your comment. As suggested, I have added more explanation and reasons for focusing on competition in the introduction. Please see the revised between line no 41 – 66.

Providing incremental contribution to the existing literature: I think that one of the strengths of this study is dealing with the unique dataset from Thailand. I hope that it provides some empirical evidence from the collected Thai dataset and more importantly provide additional explanation which can be used in understanding the Thailand Banking industry beyond and above the earlier findings from other countries. I think that it should be considered as the main point of the study.

Response: According to your suggestion, I have increased the more empirical evidences by using the unique data set for the better understanding of the sector. Please see the revised between line no 488 – 522. I have changed some results of the structural approached as well. Please see the revised between line no.439 -456.

The manuscript needs English editing before resubmitting.

Response: Thank you very much for your concern, I have proofed English with MDPI before submitting.

I again thank you for giving this opportunity to learn from your research project and I wish you the very best.

Response: Thank you very much for your advantageous feedback. Your comments are very helpful and the manuscript is much improved.

Best Regards, 

Jirawan P.

Round 2

Reviewer 1 Report

A number of my comments have not yet been addressed:

1. "It would have been useful to provide the actual data on market shares in the sector." This still has not been done.

2. "The discussion of the results for the CR5 and HHI indices concludes that the values move in opposite directions and have opposite implications, but this was due to a one-off event in 2004, the reason for which should be explained." Not yet done. The additional information in the second draft doesn't shed much light on this.

3. "The implications of the LI computation are not clear, since the trends of the SFA and FE are in the opposite direction, and it is merely concluded that the values are "in the theoretical range." So it is not clear what the point of the exercise is." The author argues that the LI (SFA) measure is preferable because of its high correlation with measures such as profitability capitalization, diversification and cost efficiency. But a high correlation is not surprising, since profitability is related to the formula for calculating LI(SFA). Also, these are effectively one-variable regressions, and so not convincing. And no particular interpretration of the meaning of the range of values of LI(SFA) is given.

Other minor comments:

lines 548-552: "In case of Thailand, the outcome of CR5 and HHI presents a very high level of concentration in banking market. Therefore, the bank’s profitability should increase. However, when we consider the Net Interest Margin: NIM (see figure 3), which is a proxy for reflecting bank’s profitability. It is obvious that
552 profitability of all banks tend to declined which contradicts the SPC theory." This argument confuses level with change. High concentration might imply high profitability, but not necessarily increasing profitability.

lines 611-613: "The average value of the LI (see Table 3) obtained from the SFA method is 0.40, and it varies from 0.36 to 0.60. The average value using the traditional method is relatively close to
that of the SFA method at 0.37, while the range is quite wide, −1.27 to 1.74." Should be Table 5, and FE figures in table are different.

Author Response

Dear Sir,

Thank you very much for your comments. Please see the attachment.

Best regards,

Jirawan P.

Reviewer 2 Report

Journal: Economies (ISSN 2227-7099)

Manuscript ID: economies-740356

Title: Empirical Measurement of Competition in the Thai Banking Industry

Review Report

I thank you for giving me the opportunity to read your response letter and the updated manuscript. I also thank you for answering my questions that I wrote on the first review report. It is much clearly and readable. The updated version does provide additional evidence beyond and above the earlier arguments in the existing literature. I wish you the very best in publish this research paper.

Regards,

Author Response

Dear Sir,

I am grateful to you for your kindly support. Thank you very much for your positive feedback.

Best Regards,

Jirawan P.